# Comprehensive Analysis of a Novel Immune-Related Gene Signature in Lung Adenocarcinoma

**DOI:** 10.3390/jcm11206154

**Published:** 2022-10-19

**Authors:** Hongxiang Feng, Chaoyang Liang, Yuhui Shi, Deruo Liu, Jin Zhang, Zhenrong Zhang

**Affiliations:** Department of Thoracic Surgery, China-Japan Friendship Hospital, Beijing 100029, China

**Keywords:** lung adenocarcinoma, immune-related gene signature, risk model, tumor infiltration immunity, prognosis

## Abstract

Lung cancer is the major cause of cancer-related deaths around the world. Lung adenocarcinoma (LUAD), the most common subtype of lung cancer, contributed to the majority of mortalities and showed different clinical outcomes in prognosis. Tumor-infiltrated immune cells at the tumor site are associated with better survival and immunotherapy response. Thus, it is essential to further investigate the molecular mechanisms and new prognostic biomarkers of lung adenocarcinoma development and progression. In this study, a six-gene signature (CR2, FGF5, INSL4, RAET1L, AGER, and TNFRSF13C) was established to predict the prognosis of LUAD patients, as well as predictive value. The prognostic risk model was also significantly associated with the infiltration of immune cells in LUAD microenvironments. To sum up, a novel immune-related six-gene signature (CR2, FGF5, INSL4, RAET1L, AGER, and TNFRSF13C) was identified that could predict LUAD survival and is highly related to B cells and dendritic cells, which may provide a theoretical basis of personalized treatment for targeted immunotherapy.

## 1. Introduction

Lung cancer is currently the most aggressive cancer in terms of morbidity and mortality worldwide [1,2,3]. There are two main types of lung cancer, small cell lung cancer (SCLC) and non-small cell lung cancer (NSCLC). The most common type of lung cancer is non-small cell lung cancer (NSCLC), which is diagnosed in 85% of all lung cancers [4,5]. Lung adenocarcinoma (LUAD) is considered to be the most common histologic subtype of NSCLC, which accounts for approximately 40% of patients with lung cancer. Although there have been major advances in lung cancer treatment in recent decades, including surgical resection, radiotherapy, chemotherapy, and immunotherapy, the prognosis and survival of patients are still unsatisfactory [6]. Possible reasons are that patients diagnosed were in advanced stages, or that patients diagnosed early cannot receive targeted therapy yet because they do not carry the common molecular mutations (EGFR, BRAF V600E, MET, or ALK) that could be treated with targeted therapy. It is necessary to further investigate the molecular mechanisms of tumorigenesis and the development of new and reliable biomarkers to improve the survival of LUAD patients.

It is known that a tumor immune microenvironment plays significant roles in tumor development and progression [7,8,9,10,11,12]. The primary cause is the inhibition of immunosuppressive checkpoints such as CTLA4 or PD-1/PD-L1 or the breakdown in the development of cytotoxic T-cell lymphocytes (CTLs) [13,14,15]. An increasing number of studies suggest that several tumor prognostic and predictive biomarkers are highly associated with immune response. For example, an immune-related gene signature of 11 genes was able to predict the prognosis and the immunotherapy efficacy of hepatocellular carcinoma (HCC) [16]. A robust six immune-related gene signature played a role in risk stratification and overall survival in patients with lower-grade glioma [17]. An uncovered seven-gene signature was verified to predict the prognosis in osteosarcoma [18]. A four-gene signature was developed to predict the prognosis and survival of lung adenocarcinoma patients [19]. These findings have been proven to demonstrate the importance of tumor immune microenvironments in carcinogenesis, progressions, and the development of tumors.

Although many immune-related gene-based markers for LUAD are available, immunotherapy could only benefit a small proportion of patients [19,20]. There is an urgent need for a more comprehensive and reliable indicator that can predict both the survival of LUAD patients and the efficacy of immunotherapy. In this study, we constructed an immune-related prognostic gene signature. Furthermore, we also performed the evaluation of prognostic significance, Gene Set Enrichment Analysis (GSEA) and the possible predictive value in immunotherapy of the gene signatures. The workflow of this study was summarized in Figure 1. We hope the immune-related prognostic gene signature could be used as the predictive biomarker for the prognosis and immunotherapy of LUAD patients.

## 2. Materials and Methods

### 2.1. Generation of Differentially Expressed Immune-Related Genes (DEIRGs)

Differentially expressed genes (DEGs) between LUAD tumor samples at stage T1 (*n* = 175) and stage T2-T4 (*n* = 357) from a TCGA dataset were assessed by using R package “DESeq2 (v 1.26.0)” with the criteria of |log2 (Fold change)| > 1 and p.adj < 0.05 [21]. The volcano plot of DEGs was drawn by using R package “ggplot2 (v3.3.3)”. The 1793 immune-related genes (IRGs) were downloaded from the Immunology Database and Analysis Portal (ImmPort) database (https://www.immport.org/home) [22]. The differentially expressed immune-related genes (DEIRGs) between DEGs and IRGs were analyzed by using R package “ggplot2 (v3.3.3)” and visualized as the Venn diagram.

### 2.2. Functional Enrichment Analyses

The 40 DEIRGs were obtained from the intersection between DEGs and IRGs. The Gene Ontology (GO) and Kyoto Encyclopedia of Genes and Genomes (KEGG) analyses were performed using clusterProfiler (v3.14.3) and org.Hs.eg.db (3.10.0) [23]. The bubble diagram was visualized by R package “ggplot2 (v3.3.3)”. Gene set enrichment analysis (GSEA) analysis was conducted by clusterProfiler (3.14.3) and shown as the mountain plot. The reference gene set was c2.cp.v7.2.symbols.gmt [Curated]. Gene set databases are MSigDB Collections. The False Discovery Rate (FDR) < 0.25 and p.Adjust < 0.05 are considered to be significant [23,24].

### 2.3. Establishment and Validation of the Prognostic Immune-Related Gene Signature

Differentially expressed immune-related genes (DEIRGs) were used for the Least Absolute Shrinkage and Selection Operator (LASSO) coefficient screen using the R software glmnet (v4.1-2) and survival (3.2-10). The prognostic immune-related gene signature (risk model) with coefficients were selected based on the optimal lambda value. Overall survival (OS), disease specific survival (DSS) and the progression-free interval (PFI) between high and low risk score groups were conducted using the R software survminer (0.4.9) and survival (3.2-10) [24]. Univariate and multivariate Cox analyses and 1-, 3-, and 5-year receiver operating characteristic (ROC) analysis were used to estimate the prognostic value of the risk model [25]. The predictive risk factors were evaluated by Nomogram and Calibrate curves using the R software rms (6.2-0) and survival (3.2-10) [25].

### 2.4. Immune Cell Infiltration Analysis

The ssGSEA algorithm was applied for the correlation between the risk score and 24 tumor-infiltrated immune cells by using GSVA (1.34.0) [26,27]. The ESTIMATE method was used for the comparison of the immune score, ESTIMATE score, and stromal score between the low- and high-risk groups [28]. The prognostic value of the 24 tumor-infiltrated immune cells was analyzed by the univariate Cox regression analysis and Kaplan–Meier analysis.

### 2.5. Gene Expression and Genetic Alterations Analysis

The immunohistochemistry images of six gene signatures in LUAD tumor tissues and normal tissues were obtained from the Human Protein Atlas (HPA) database (https://www.proteinatlas.org). The Genetic alteration analysis was performed using the cBioPortal for Cancer Genomics (https://www.cbioportal.org) [29].

### 2.6. Reverse Transcription and Real-Time Quantitative PCR (RT-qPCR)

Human lung cancer cell lines (A549, NCI-H1975, Calu-3) and human normal epithelial cells BEAS-2B were purchased from Procell (https://www.procell.com.cn). The A549 cells were cultured in Ham’s F-12K Medium (Thermo Fisher, Waltham, MA, USA). The NCI-H1975 cells were cultured in RPMI1640 medium (Hyclone, Logan, UT, USA). The Calu-3 cells were cultured in modified Eagle’s medium (MEM, containing NEAA, Procell, Wuhan, China). The BEAS-2B cells were cultured in Dulbecco’s modified Eagle’s medium (DMEM, HyClone, USA). All cells were cultured in the specific mediums supplemented with 10% fetal bovine serum (Hyclone, USA), penicillin sodium (100 U/mL) and streptomycin (100 mg/mL). The cells were incubated in the thermostatic cell incubator at 37 °C with 5% CO_2_. Total RNA was extracted using the RNeasy Plus Mini Kit (QIAGEN, Hilden, Germany). Quantitative PCR (qPCR) was performed using the ABI 7900 qPCR system. Relative RNA expression was normalized to GAPDH and calculated by the 2^−ΔΔCt^ method. The primers were listed in Table 1.

### 2.7. Statistical Analysis

Data were analyzed using the R software (v3.6.3) and the qPCR analysis was performed by GraphPad Prism. DEGs were analyzed by “ggplot2”. GO-KEGG and GSEA analyses were performed by “clusterProfiler”. The LASSO analysis was conducted by “glmnet and survival”. Univariate and multivariate Cox regression analyses were assessed by using the “survival”. The Kaplan–Meier analysis was performed using the “survival” and “survminer” R packages. An immune cell infiltration analysis was applied by “GSVA”. A *p* value < 0.05 is statistically significant.

## 3. Results

### 3.1. Identification of DEIRGs

The workflow of this study is illustrated in Figure 1. We analyzed a total of 535 LUAD samples and 59 normal samples and obtained 697 DEGs, including 411 upregulated and 286 downregulated genes as shown in the volcano map (Figure 2A). In addition, the forty differentially expressed immune-related genes (DEIRGs) were generated (Figure 2B). The functional analysis of the DEGs was performed. The GO enrichment analysis showed that the DEGs were enriched in “humoral immune response”, “defense response to bacterium”, and “antimicrobial humoral response” (Figure 2C). The KEGG pathway analysis revealed that the highly relevant signaling pathways were neuroactive ligand—receptor interaction and complement and coagulation cascades (Figure 2D). The functional analysis (GOs and KEGGs) for the remaining genes that were differentially regulated in cancer versus normal tissue were also performed. The enriched functions of either upregulated or downregulated genes were not related to immune responses as shown in Figure 2E,F.

### 3.2. Construction of a Prognostic Immune-Related Gene Signature

We used a LASSO Cox regression model to perform the overall survival analysis of 40 immune-related potential prognostic genes (Figure 3A). Six genes included in the analysis were identified to be the best predictors for prognostic significance (Figure 3B). Hence, the prognostic model consists of these six genes noted as CR2, FGF5, INSL4, RAET1L, AGER, and TNFRSF13C. The regression coefficient of each gene was displayed in Table 2. The risk scores of this model are formulated as follows: Risk score = −0.01996 * expression level of CR2 + 0.07140 * expression level of INSL4 + 0.03141 * expression level of FGF5 + 0.12065 * expression level of RAET1L + (−0.00451) * expression level of AGER + (−0.00301) * expression level of TNFRSF13B.

### 3.3. Evaluation of the Prognostic Value of the Six-Gene Signature

The risk score of each LUAD patient was a linear combination of each six-gene signature expression and its risk coefficient. High- and low-risk groups of LUAD patients were divided according to the median risk score. The distribution plots of risk scores and the outcome status, including the Overall Survival (OS), Disease Specific Survival (DSS), and Progression-Free Interval (PFI) of the gene signature, are shown in Figure 4A. Patients in the high-risk group showed a lower survival probability compared with those in the low-risk group. Consistently, Kaplan–Meier plotter analysis showed that the OS, DSS and PFIs of the patients in the high-risk group exhibited remarkably worse outcomes than those in the corresponding low-risk group (Figure 4B). Next, we performed the univariate analysis to validate the six immune-related genes with prognosis significance in LUAD (Figure 4C). In addition, we generated a nomogram tool to predict the 1-year, 3-year and 5-year overall survival risk based on the selected factors (Figure 4D). Moreover, the calibration curves for the prognostic nomograms presented a better accuracy and consistency between prediction and actual 1-, 3-, and 5-year survival in the TCGA-LUAD cohort (Figure 4E).

We performed the time-dependent receiver operating characteristic (ROC) plotter and found that the values of the area under the curve (AUC) for the six-gene signature risk score at 1, 3, 5, and 10 years were all above 0.6 (Figure 5A). Next, we performed the correlation between the six genes individually and the risk score analysis using the Pearson correlation analysis as shown in Figure 5B–H. TNFRSF13B was significantly related with most of the genes, except for RAET1L, among the six-gene signature model (Figure 5B). There were negative correlations between CR2, AGER, TNFRSF13B and the risk score (Figure 5C,E,H). There was a positive relationship between RAET1L, FGF5, INSL4 and the risk score (Figure 5D,F,G). These results indicated that the six-gene signature risk score could be used as an independent prognostic factor for LUAD.

### 3.4. Functional Enrichment Analysis of the Six-Gene Signature Risk Score

To further understand the biological functions of the six-gene signature risk score in LUAD, we performed enrichment analyses of GO and KEGG pathways. Results showed that the six-gene signature was enriched in lymphocyte mediated immunity, leukocyte proliferation, mononuclear cell proliferation, and lymphocyte proliferation (Figure 6A). Consistently, multiple GSEA analyses revealed that the six-gene signature was highly enriched in immune-related signaling pathways such as antigen activates B cell receptor (BCR) leading to the generation of second messengers, reactome immunoregulatory interactions between a lymphoid and a non-lymphoid cell, reactome interleukin 20 family signaling, type II interferon signaling, and CD22 mediated BCR regulation (Figure 6B–G).

### 3.5. Clinicopathological Characteristics Correlation Analysis

Next, we investigated the possible relationship between multiple clinicopathological features and the six-gene signature risk score. The correlation analysis indicated that a higher risk score was significantly associated with age, gender, pathologic stages, T/N/M stages, and OS/DSS/PFI events (Figure 7A–I). Moreover, we also did univariate and multivariate Cox analyses with risk scores as shown in Table 3. The results validated that higher age, T/N/M and pathologic stage, as well as higher risk scores were risk factors for LUAD patients with HRs > 1, *p* < 0.01.

### 3.6. Evaluation of Immune Cell Infiltration Characterization

To further explore the association between the six-gene signature risk score and immunity, we firstly evaluated the correlation with 24 types of tumor-infiltrating immune cells in the high- or low-risk groups using the ssGSEA algorithm (Figure 8A). Results showed that T cells, plasmacytoid DC (pDC), mast cells, DC, cytotoxic cells, B cells, and Th17 cells showed a lower expression in the high-risk group (*p* < 0.05), whereas T gamma delta (Tgd) and Th2 cells were highly expressed in the high-risk group. Furthermore, we investigated the correlation with tumor lymph cells by Spearman’s correlation analysis. The lollipop chart indicated that risk score was positively related to Th2 cells and Tgd (Figure 8B). Moreover, a negative correlation was observed between risk score and TFH, B cells, Mast cells, T cells, and CD8 T cells (Figure 8B).

In addition, we also investigated the prognostic significance of 22 types of tumor-infiltrating immune cells. The Kaplan–Meier analysis demonstrated that the lower infiltrating abundance of B cells and dendritic cells were significantly associated with better OS (*p* < 0.05, Figure 8C,D). The univariate Cox analysis revealed that B cells and CD4 T cells remarkably influenced prognosis (Figure 8E). These findings indicate that B cells and CD4 T cells may play a meaningful role in the prognostic ability of the six-gene signature in LUAD.

### 3.7. Estimation of the Immunity Response

We also evaluated the association between the six-gene signature and tumor immune infiltration to calculate the immune score, ESTIMATE score, and stromal score using the ESTIMATE package. We observed that there was a significant difference in the immune score and stromal score between the low-risk group and the high-risk group, except for the ESTIMATE score (Figure 9A–C), suggesting that the six-gene signature risk score was significantly associated with the immune score and stromal score. Next, we also did the correlation analysis of immune checkpoints as shown in Figure 9D–I. There was a significant relationship between CD27, TNFSF15, TNFRSF25, TIGIT, VSIR, BTNL2 and risk score, suggesting that the six-gene signature risk score might have an immune response and respond to immunotherapy in LUAD.

### 3.8. Verification of Hub Genes

The mRNA level of CR2, INSL4, FGF5, RAET1L and TNFRSF13B was upregulated in LUAD tumor tissues compared with paired normal tissues whereas AGER was downregulated (Figure 10A). The protein distribution from the HPA database showed the consistency at protein level of CR2, TNFRSF13B, and AGER in LUAD tissues without the other three other genes present for the immunohistochemistry staining as they were not available in the HPA (Figure 10B). In addition, we also measured the mRNA levels of the six hub genes in LUAD cell lines, including A549, NCI-H1975, and Calu-3 as shown in Figure 10C, of which the patterns were consistent with the RNA level of tumor tissues. We also analyzed genetic alterations of the hub genes in LUAD via cBioportal. There were 16% (186/1144) genetic alterations in total, and CR2 was found to have the highest alteration (8%) in LUAD patients (Figure 10D).

## 4. Discussions

In recent years, the development of cancer treatment strategies has been toward immunotherapy [30,31]. Cell-mediated immunity has been shown to play a role in both lung cancer monitoring and development. As studies on first-line immunotherapy for NSCLC proceeds, it is more and more challenging to choose the most suitable therapeutic treatment for patients with different clinicopathological features from among such options as immunotherapy alone, or chemotherapy combined with immunotherapy, or immunotherapy combined with other treatment [32]. Thus, it is essential and urgent to develop an immune-related prognostic model to better predict the prognosis and evaluate the efficacy of immunotherapy in LUAD patients. In this study, we identified 40 significantly different immune- related genes between LUAD tumors and normal tissues. We established a prognostic risk model that included six hub genes as determined by LASSO, univariate, and multivariate regression analyses. Moreover, the gene signature was found to be highly associated with tumor-infiltrated immune cells, which might be used as an independent factor. The prognostic risk model consisting of CR2, INSL4, FGF5, RAET1L, AGER, and TNFRSF13C acts as a novel potential biomarker for evaluating the prognosis and the efficacy of immunotherapy in LUAD.

Nowadays, it has become a growing focus for research to construct prognostic models based on lncRNA, miRNA, and mRNA to predict the prognostic or diagnostic value in tumors [33]. In this study, the gene signature was established by taking the cross from LUAD transcriptome differential expressed genes and immune-related genes. We found a total of 40 genes that were involved in the “humoral immune response” and “defense response to bacterium”, suggesting that the gene signature was closely related to immune response. Next, LASSO and regression analyses were performed to identify 6 out of these 40 immune-related genes to construct a prognostic risk model, which included CR2, INSL4, FGF5, RAET1L, AGER, and TNFRSF13C. Therein, the complement receptor type 2 (CR2) was reported to combine with the B-Cell receptor to inhibit the activation, proliferation, and antibody production of human B cells [34]. CR2 was also verified to be involved in the nine-gene signature with the prognostic and predicted values for LUAD [35]. Insulin-like 4 (INSL4) has been reported to be required for the growth and viability of LKB1-inactivated lung cancer [36]. A recent study showed that INSL4 might be a prognostic marker for proliferation and invasiveness in non-small cell lung cancer (NSCLC) [37]. It was reported that FGF5 could promote cell proliferation via activation of the MAPK signaling pathway in osteosarcoma [38]. Moreover, high FGF5 expression was found to be significantly associated with poor overall survival and relapse-free survival in LUAD [39]. A report showed that RAET1L together with ULBP1, ULBP2, ULBP3 had the high diagnostic values in colon adenocarcinoma (COAD) [40]. AGER overexpression was found to suppress the cell proliferation, invasion and migration of H1299 cells [41]. TNFRSF13C has been demonstrated to be involved in a five-gene signature (CD40LG, TNFRSF6B, TNFSF13, TNFRSF13C, and TNFRSF19) which was helpful for the prognosis and immunotherapy response prediction in LUAD [42]. These previous studies have revealed that several genes were closely related to lung adenocarcinoma, which also further confirmed the validity and reliability of our prognostic six-gene risk model.

Studies have shown that a tumor microenvironment plays a key role in NSCLC development and is closely related to immunotherapy response [43,44]. Given the importance of tumor immune infiltration, we performed the correlation analysis of immune infiltrates using ssGSEA and stromal score algorithms. Recent studies have shown that Th2 cells have tumor-promoting effects in lung cancer and even human primary NSCLC tumors [45,46,47]. Existing evidence shows that tumor-infiltrating B cells play a role in almost all stages of lung cancer [48,49]. Consistent with the above findings, our work showed that the risk score (six-gene signature) was highly correlated with Th2 and B cells, suggesting a potential role in immune response. Moreover, we identified that higher infiltrations of B cells and dendritic cells were associated with a better cumulative survival of LUAD patients, findings which were well in line with recent research showing that a gene expression signature associated with B cells was correlated with survival using immunotherapy in lung adenocarcinoma [50]. Our findings were identical to a clinical trial where personalized neoantigen pulsed dendritic cell vaccines were administered for advanced lung cancer [51]. Above all, the six-gene signature risk model constructed in this study may have a better prognosis value and clinical significance in LUAD. Furthermore, we built a nomogram to assess the prediction accuracy by linking the risk model with T stage, N stage, age, and gender. Similarly, it was reported that a nomogram was also used to enhance the prognostic prediction accuracy by combining the risk index in HCC patients with clinicopathological features [16]. Thus, it is of significance to provide and improve the reliability and efficacy of survival risk prediction for LUAD patients by using the novel prognostic model.

The present study identified an original immune-related risk model for LUAD. There are some highlights in this study. First, we did a comprehensive analysis of the six-gene signature in LUAD, including the functional enrichment, the correlation with clinical features, the tumor-infiltrated immune cells, the immune checkpoints, and the expression profiles in tissues and cell lines. Second, a nomogram and calibration were used to further validate the accuracy for clinical outcomes for the risk model in LUAD. Nevertheless, there are some limitations in this study. First, clinical LUAD samples need to be collected to verify the validity and accuracy of the prognostic model. Second, relevant functional and mechanical experiments should be carried out to further explore the effect of this risk model in vitro and in vivo.

## 5. Conclusions

In conclusion, we established an immune-related prognostic gene signature which was significantly associated with the tumor immune microenvironment. The six-gene signature model could predict the prognosis and survival of LUAD patients, as well as imply a better response to immune-based therapies, which may provide a theoretical basis for the prognosis and immunotherapy of LUAD.

## Figures and Tables

**Figure 1 jcm-11-06154-f001:**
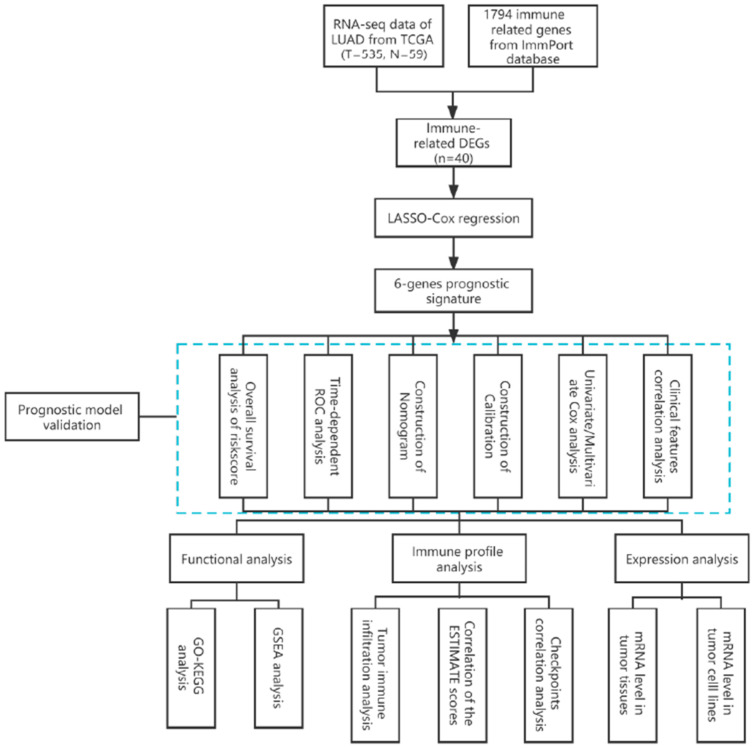
The workflow of this study.

**Figure 2 jcm-11-06154-f002:**
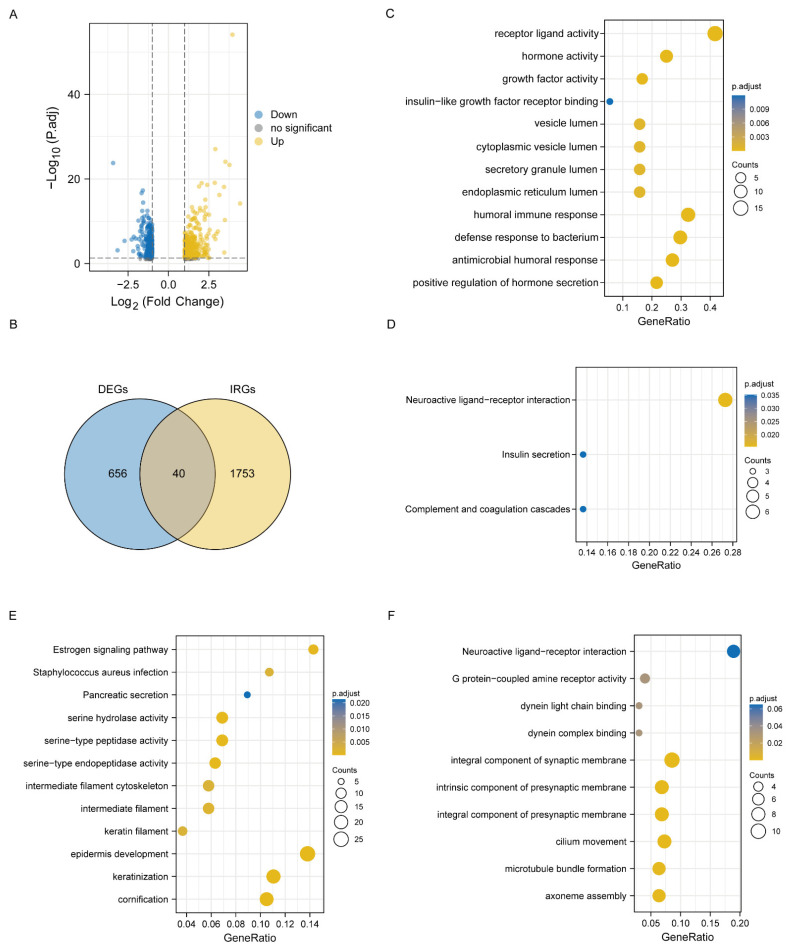
Construction of DEIRGs for LUAD. (**A**) Volcano plot of DEIRGs between LUAD and normal tissues. (**B**) Venn diagram of the intersections of DEGs with IRGs. (**C**) GO enrichment analysis of forty DEIRGs. (**D**) KEGG enrichment analysis of forty DEIRGs. (**E**) GO and KEGG enrichment analysis of the upregulated remaining genes (except the above forty genes). (**F**) GO and KEGG enrichment analysis of the downregulated remaining genes (except the above forty genes).

**Figure 3 jcm-11-06154-f003:**
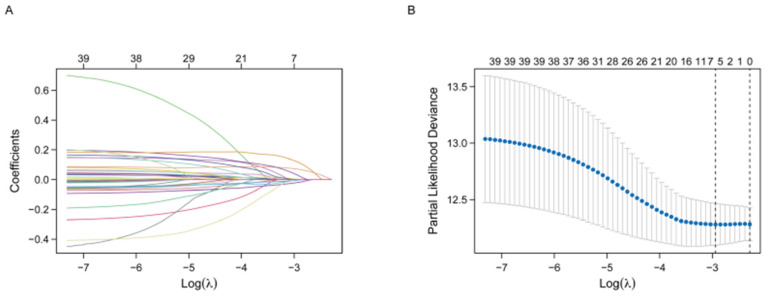
Establishment of the prognostic gene signature using LASSO regression analysis. (**A**) LASSO coefficient profiles of 40 immune-related potential prognostic genes. Each curve corresponds to a gene. (**B**) Tenfold cross-validation for tuning parameter selection in the LASSO model. The partial likelihood deviance is plotted against log (Lambda), where Lambda is the tuning parameter. Partial likelihood deviance values are shown, with error bars representing SE. The dotted vertical lines are drawn at the optimal values by minimum criteria and 1-SE criteria. LASSO—the least absolute shrinkage and selection operator Cox regression model.

**Figure 4 jcm-11-06154-f004:**
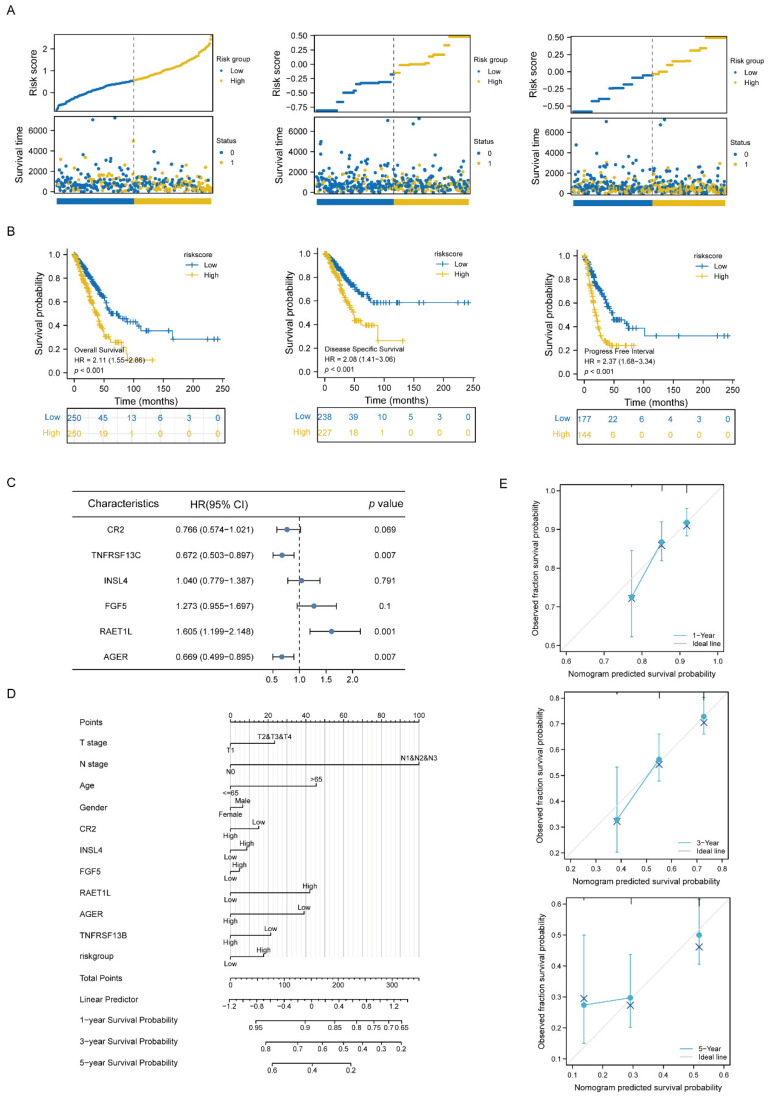
Evaluation of prognostic significance of the six-gene signature risk score. (**A**) Risk score distribution and outcome status, including OS, DSS, and PFI of LUAD patients in low- and high-risk groups. (**B**) Kaplan–Meier plotter curves of the six-gene signature risk score. (**C**) Forest plot for the multivariable Cox model results of each gene in the six-gene signature risk score. (**D**) Nomogram for predicting 1-, 3-, and 5-year OS in TCGA-LUAD cohort. (**E**) Calibration curves of prognostic nomogram on consistency between prediction and observed 1-, 3-, and 5-year survival in TCGA-LUAD cohort. Dashed line at 45° implicated a perfect prediction, and the actual performances of our nomogram were shown in blue lines. OS—overall survival. DSS—disease specific survival. PFI—progression-free interval.

**Figure 5 jcm-11-06154-f005:**
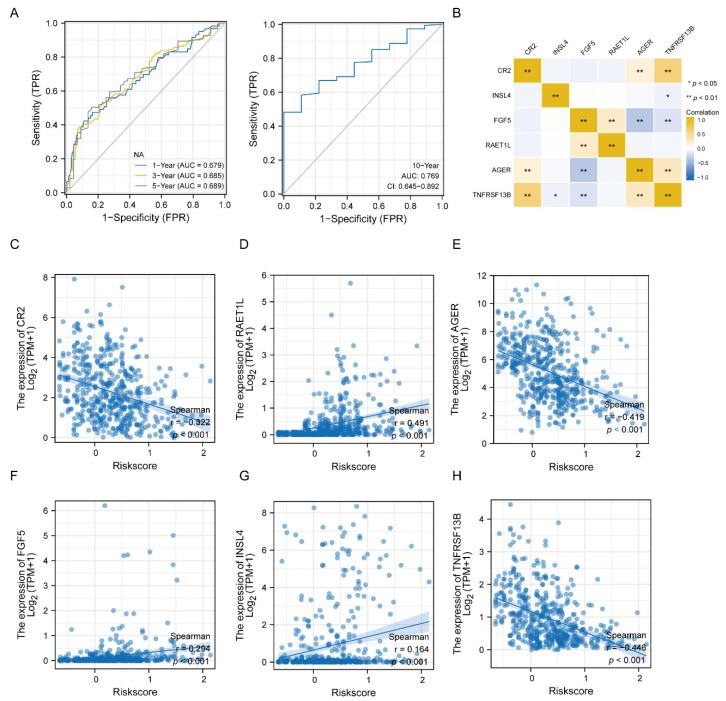
ROC analysis and molecular correlation analysis of the six-gene signature risk score. (**A**) The 1-year, 3-year, 5-year and 10-year ROC curve plot of LUAD patients. (**B**) Heatmap shows the correlation between the six-gene signature molecules. (**C**–**H**) Scatter diagram shows correlation between risk score and CR2 (**C**), RAET1L (**D**), AGER (**E**), FGF5 (**F**), INSL4 (**G**), and TNFRSF13B (**H**). *p* < 0.001. ROC—receiver operating characteristic.

**Figure 6 jcm-11-06154-f006:**
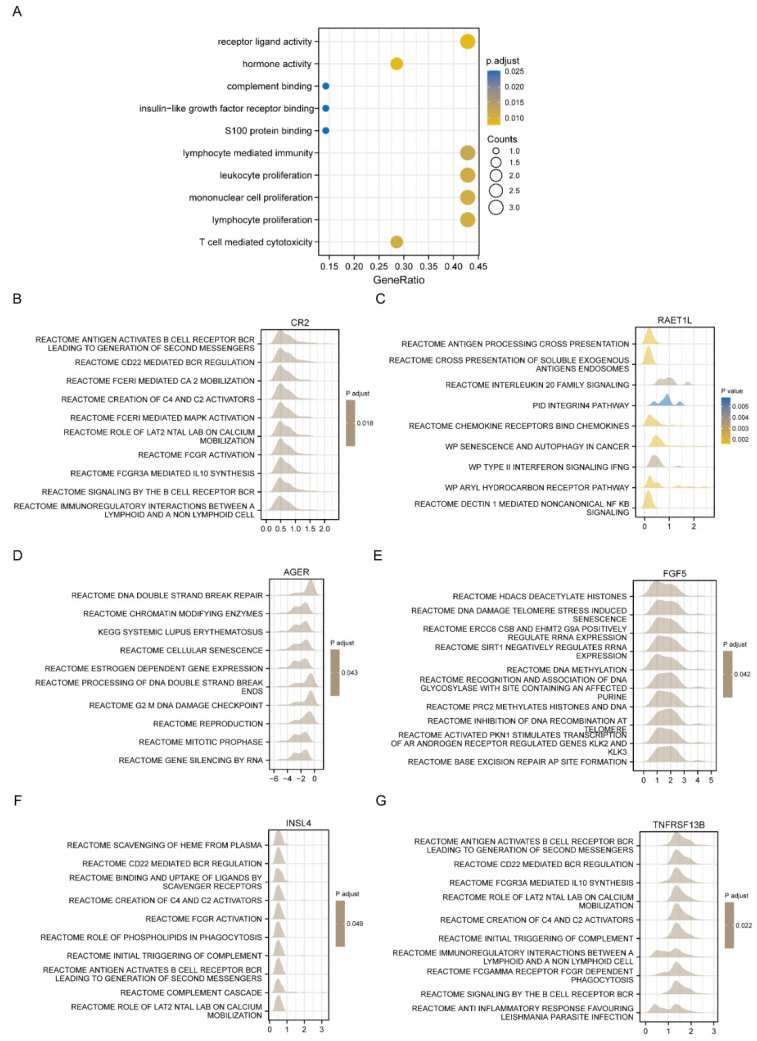
Functional enrichment analysis of six-gene signature. (**A**) GO and KEGG pathway analysis indicating the enriched pathways with six-gene signature risk score. (**B**–**G**) GSEA analysis of differential genes associated with CR2, RAET1L, AGER, FGF5, INSL4, and TNFRSF13B, respectively.

**Figure 7 jcm-11-06154-f007:**
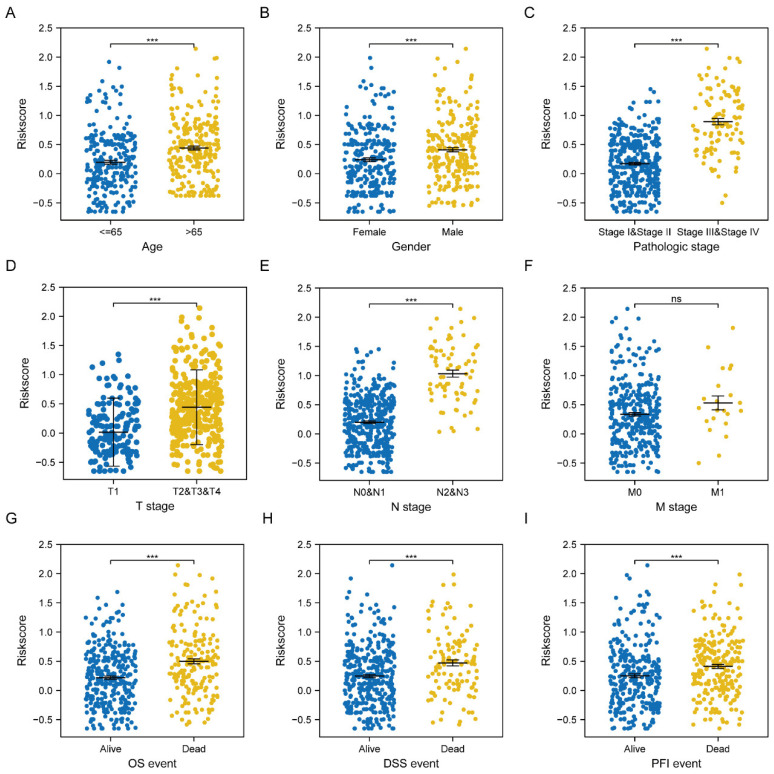
Clinicopathological features correlation analysis. (**A**–**I**) The relationships between risk score and clinicopathological factors (Age, Gender, Pathologic stage, T stage, N stage, M stage, OS, DSS, and PFI events) in TCGA-LUAD cohort, respectively. *** *p* < 0.001.

**Figure 8 jcm-11-06154-f008:**
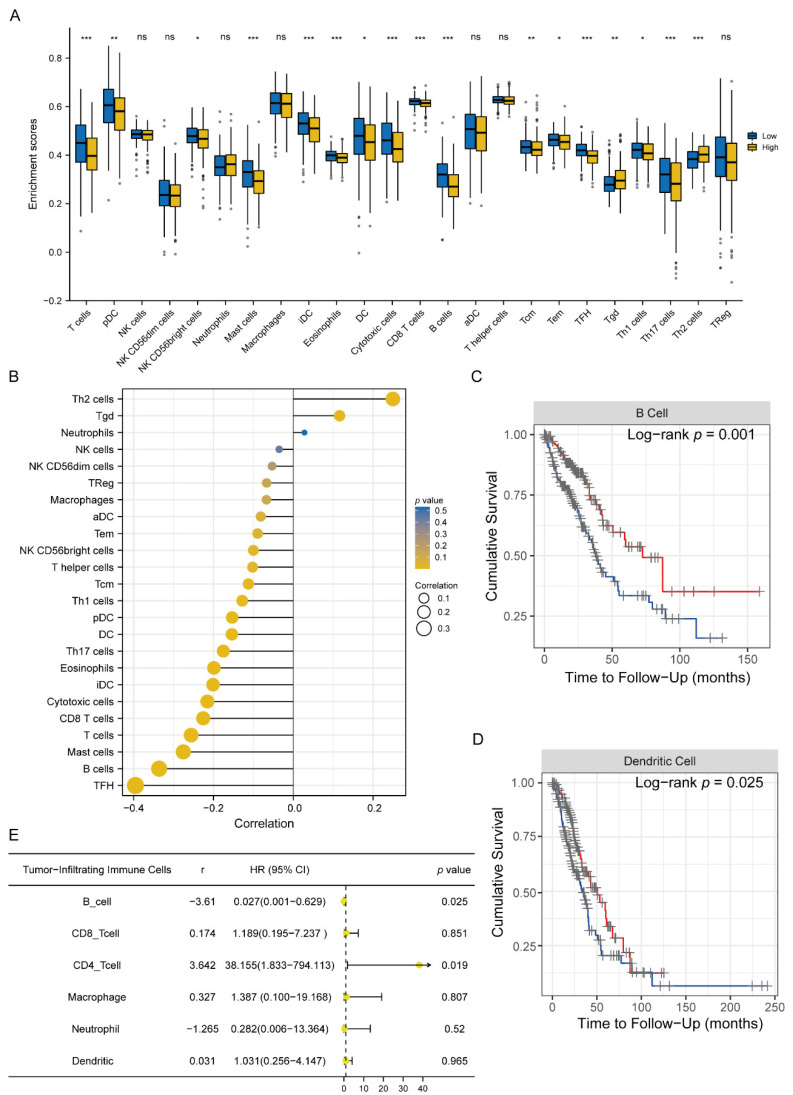
Relationship between immune cell infiltration and six-gene signature risk score and prognostic value evaluation. (**A**) Correlations of risk score with immune cell infiltration. The blue and yellow boxes represented the six-gene signature low- and high-risk group, respectively. The medium line inside the box represented the mean value. Wilcoxon rank-sum was applied for the significance test. * *p* < 0.05; ** *p* < 0.01; *** *p* < 0.001. (**B**) Lollipop chart showing the Spearman analysis of risk score with tumor-infiltrating immune cells. (**C**) Relationship between overall survival and B cells. High level (Red line) vs. low level (black line). (**D**) Relationship between overall survival and dendritic cells. High level (Red line) vs. low level (black line). (**E**) Univariate Cox regression model built for six tumor-infiltrating immune cell types (B cells, CD8 cells, CD4 cells, Macrophage, Neutrophil, Dendritic) based on overall survival. T cells; pDC [Plasmacytoid DC]; NK cells; NK CD56dim cells; NK CD56bright cells; Neutrophils; Mast cells; Macrophages; iDC [immature DC]; Eosinophils; DC; Cytotoxic cells; CD8 T cells; B cells; aDC [activated DC]; T helper cells; Tcm [T central memory]; Tem [T effector memory]; Tfh [T follicular helper]; Tgd [T gamma delta]; Th1 cells; Th17 cells; Th2 cells; and Treg.

**Figure 9 jcm-11-06154-f009:**
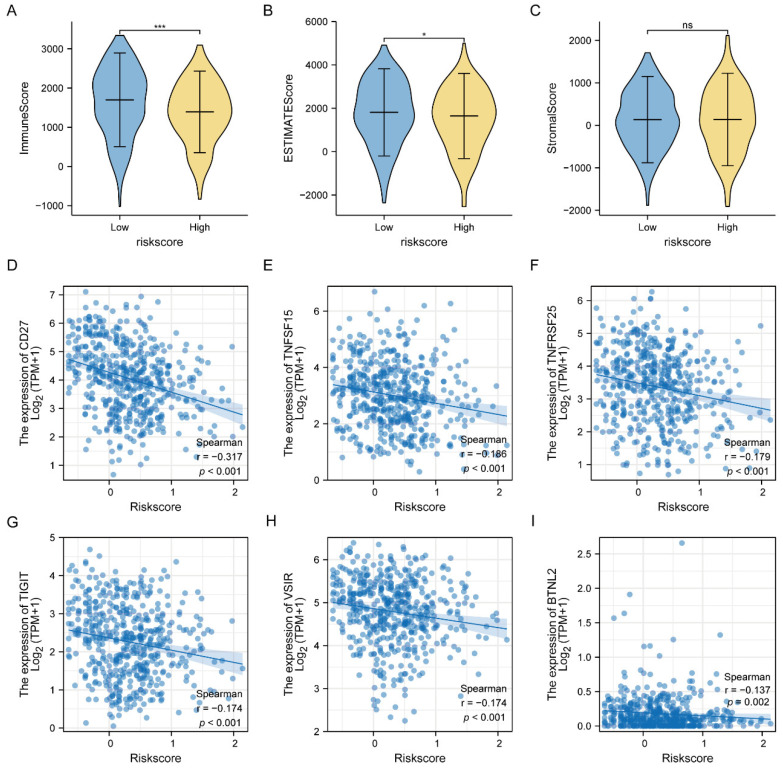
Correlation analysis of six-gene signature risk score with the ESTIMATE score and immune checkpoints. (**A**–**C**) Immune score, ESTIMATE score, and stromal score between low risk-group and high-risk group of six-gene signature, respectively. * *p* < 0.05; *** *p* < 0.001. (**D**–**I**) Correlation analysis of immune checkpoints (CD27, TNFSF15, TNFRSF25, TIGIT, VSIR, and BTNL2) with risk score, respectively. *p* < 0.001.

**Figure 10 jcm-11-06154-f010:**
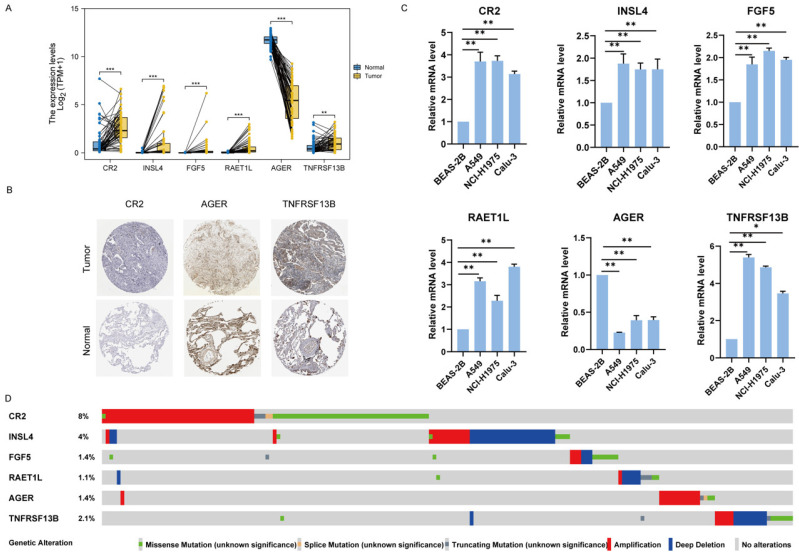
Verification of the expression and genetic alterations of 6 hub genes in tumor and normal tissues. (**A**) The boxplot showed the expression of 6 hub genes between paired LUAD tumor and normal tissues in TCGA database. (**B**) The immunohistochemical staining of CR2, AFER, TNFRSF13B was obtained from HPA database. (**C**) The mRNA levels of hub genes were measured by qRT-PCR in lung adenocarcinoma cell lines (A549, NCI-H1975, and Calu-3) and compared with human normal lung epithelial cells BEAS-2B. (**D**) Genetic alterations of hub genes in LUAD in TCGA datasets. * *p* < 0.05, ** *p* < 0.01, *** *p* < 0.001.

**Table 1 jcm-11-06154-t001:** Primers for qRT-PCR.

Gene	Forward Sequence (5′-3′)	Reverse Sequence (5′-3′)
CR2	ACCATGGTCGTCATACAGGTG	AGCCAGGATTGCATCAACA
INSL4	AGCCTGTTCCGGTCCTATCT	ATGATGGCTGCCCTTCAGAC
FGF5	CGCTCACAGTCACCTGGTTT	CACCCTCGTTTGGCTTTTCC
RAET1L	CCATCCCAGCTTTGCTTCTGT	TGACGGGTGTGACTGTCTTG
AGER	GCTTGGAAGGTCCTGTCTCC	CCACCAATTGGACCTCCTCC
TNFRSF13B	GTCAAAGTCCGGCCAAGTCT	CCACTGTCTGGGATGTGTGG
GAPDH	GAGAAGGCTGGGGCTCATTT	AGTGATGGCATGGACTGTGG

**Table 2 jcm-11-06154-t002:** Six immune-related prognostic genes obtained from LASSO Cox regression model.

Gene Symbol	Description	Risk Coefficient
CR2	Complement C3d receptor 2	−0.019957508
INSL4	Insulin-like 4	0.071400283
FGF5	Fibroblast growth factor 5	0.031415586
RAET1L	Retinoic acid early transcript 1L	0.120647299
AGER	Advanced glycosylation end-product specific receptor	−0.004511221
TNFRSF13B	TNF receptor superfamily member 13B	−0.003010621

**Table 3 jcm-11-06154-t003:** Univariate and multivariate Cox regression analyses of risk score and other clinicopathologic factors for OS in the entire TCGA cohort.

Characteristics	Total (N)	Univariate Analysis		Multivariate Analysis	
		Hazard Ratio (95% CI)	*p* Value	Hazard Ratio (95% CI)	*p* Value
T stage	501				
T1	168	Reference			
T2&T3&T4	333	1.668 (1.184–2.349)	0.003	1.591 (1.004–2.520)	0.048
N stage	492				
N0	325	Reference			
N1&N2&N3	167	2.606 (1.939–3.503)	<0.001	1.790 (1.004–3.191)	0.048
M stage	360				
M0	335	Reference			
M1	25	2.111 (1.232–3.616)	0.007	1.371 (0.744–2.523)	0.311
Gender	504				
Female	270	Reference			
Male	234	1.060 (0.792–1.418)	0.694		
Age	494				
≤65	238	Reference			
>65	256	1.228 (0.915–1.649)	0.171		
Pathologic stage	496				
Stage I	270	Reference			
Stage II & Stage III & Stage IV	226	2.975 (2.188–4.045)	<0.001	1.177 (0.617–2.244)	0.621
Smoker	490				
No	71	Reference			
Yes	419	0.887 (0.587–1.339)	0.568		
Risk group	482				
Low	244	Reference			
High	238	2.217 (1.625–3.025)	<0.001	1.612 (1.081–2.402)	0.019

## Data Availability

The datasets used in the current study are available, and can be found in the article.

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
