# Peer review of "Comprehensive Analysis of a Novel Immune-Related Gene Signature in Lung Adenocarcinoma"

_jcm, 2022, doi:10.3390/jcm11206154_

Round 1

Reviewer 1 Report

Lung cancer is one of the most aggressive and deadliest cancer types and causes about 1.4 × 106 global deaths every year. The development of cancer treatment strategies based on immuno-therapy give big hope towards the increased overall response rate outcomes for Lung cancer patients. However, choosing the suitable therapeutic treatment for patients with different clinicopathological features still represents a big challenge at the moment. One of the crucial aspects to reach a higher success rate on cancer treatment is to develop a robust immune-related prognostic approach. In their manuscript Hongxiang Feng et al. aim to identify markers that would help to predict the prognosis and evaluate the immunotherapy efficiency in the treatment of lung cancer patients. Starting from the analysis of differentially expressed genes between LUAD tumor samples at stage T1 and stage T2-T4 and their overlap with a dataset of immune-related genes, the authors identified a specific gene signature which might be used as a prognostic risk factor.

I feel that the manuscript from Hongxiang Feng et al. is valuable and in general it addresses a relevant topic that I believe is interesting for a broad audience.

Below are reported some comments that I feel would be beneficial to improve the manuscript.

1)    At the scientific point of view In Figure 2 the 40 genes derived by the intersection between the differentially expressed genes on LUAD tumor samples and the immune-related genes was analyzed further using GO enrichment analysis KEGG enrichment analysis. Per se doesn’t sound too surprising that genes belonging to a dataset of immune related genes will end up giving GO terms falling into the immune response. Maybe the authors would like to comment these results a bit further in order to highlight the relevance of these terms in relation to the potential therapeutic approach.

2)    Another aspect that may be interesting to address and that will add value to the study, is to try and make the same analysis (GOs and KEGGs ) also for the remaining genes that are differentially regulated in cancer versus normal tissue. Best if this is done by doing the analysis separating the upregulated and the downregulated.

3) I would strongly recommend a careful English editing of the manuscript to restructure some sentences and correct some typos throughout the text

Author Response

1. At the scientific point of view In Figure 2 the 40 genes derived by the intersection between the differentially expressed genes on LUAD tumor samples and the immune-related genes was analyzed further using GO enrichment analysis KEGG enrichment analysis. Per se doesn’t sound too surprising that genes belonging to a dataset of immune related genes will end up giving GO terms falling into the immune response. Maybe the authors would like to comment these results a bit further in order to highlight the relevance of these terms in relation to the potential therapeutic approach.

Reply: Thanks for your question. Actually we just briefly verify the immune pathways that they're enriched in and in order to implicate the potential immunotherapeutic significance. It seems that the logic of this study was more straightforward.

2. Another aspect that may be interesting to address and that will add value to the study, is to try and make the same analysis (GOs and KEGGs ) also for the remaining genes that are differentially regulated in cancer versus normal tissue. Best if this is done by doing the analysis separating the upregulated and the downregulated.

Reply: Thanks for your advice. It is actually a good idea. We did the same analysis (GOs and KEGGs ) for the remaining genes that were separated as the upregulated and the downregulated genes as shown in Figure 2E and 2F. The relevant figure legends and results description were supplemented in highlighted fonts.

3. I would strongly recommend a careful English editing of the manuscript to restructure some sentences and correct some typos throughout the text.

Reply: Thanks for your kind suggestion. We have checked though the whole text and revised some sentences, grammar and spelling mistakes as shown in highlighted fonts.

Reviewer 2 Report

This is an interesting report on a clinically important topic, which combines a thorough bioinformatic analysis with wet lab data. Main comments:

1. This score was derived from the difference between T1 and T2-T4 tumors (lines 82-83), which is prognostic (this is actually how the T-descriptor was defined, see https://pubmed.ncbi.nlm.nih.gov/26134221/), therefore it is trivial that the score is also be associated with survival (as shown in Figures 4+5). One important question is whether the proposed risk score can segregate survival better than the T-descriptor itself (T1 vs.   T2-4) or not. I suggest to add this comparative analysis to the Results. 

2.  for the Discussion: a similar gene expression signature associated with B cells has been shown to correlate with survival from immunotherapy in lung adenocarcinoma (https://pubmed.ncbi.nlm.nih.gov/33520406/), which suggests that the findings of the current study have wider importance for the biology and  prognosis of these tumors.

3. (minor) line s 46-47: adenocarcinoma is about 40% of lung cancer (not NSCLC).

Author Response

1. This score was derived from the difference between T1 and T2-T4 tumors (lines 82-83), which is prognostic (this is actually how the T-descriptor was defined, see https://pubmed.ncbi.nlm.nih.gov/26134221/), therefore it is trivial that the score is also be associated with survival (as shown in Figures 4+5). One important question is whether the proposed risk score can segregate survival better than the T-descriptor itself (T1 vs. T2-4) or not. I suggest to add this comparative analysis to the Results.

Reply: Thanks a lot for your advice. The Nomogram (Figure 4D) of riskscore with T stage (T1 vs. T2-4) for overall survival in Figure 4D and univariate and multivariate Cox regression analyses of riskscore with T stage (T1 vs. T2-4) for overall survival in Table 3 have been re-analyzed and updated in the results. The relationship between riskscore and T stage (T1 vs. T2-4) was also re-analysis and updated as shown in Figure 7D.

2.  for the Discussion: a similar gene expression signature associated with B cells has been shown to correlate with survival from immunotherapy in lung adenocarcinoma (https://pubmed.ncbi.nlm.nih.gov/33520406/), which suggests that the findings of the current study have wider importance for the biology and prognosis of these tumors.

Reply: Thanks a lot for your advice. We have cited this reference in the discussion, which could better demonstrate the significance of our established prognostic model.

3. (minor) line s 46-47: adenocarcinoma is about 40% of lung cancer (not NSCLC).

Reply: Thanks for pointing it out. we have revised “NSCLC” as “... 40% of lung cancer”.